# Performance of a Handheld Near-Infrared Spectroscopy Device to Predict Pork Primal Belly Fat Iodine Value and Loin Lean Intramuscular Fat Content

**DOI:** 10.3390/foods12081629

**Published:** 2023-04-13

**Authors:** Stephanie Lam, David Rolland, Sophie Zawadski, Xinyi Wei, Bethany Uttaro, Manuel Juárez

**Affiliations:** Lacombe Research and Development Centre, Agriculture and Agri-Food Canada, 6000 C and E Trail, Lacombe, AB T4L 1W1, Canada

**Keywords:** pork belly, loin chop, near-infrared (NIR) spectroscopy, Tellspec, iodine value

## Abstract

The increase in market demand and economic value of Canadian pork primal cuts has led to a need to assess advanced technologies capable of measuring quality traits. Fat and lean composition were measured using a Tellspec near-infrared (NIR) spectroscopy device to predict the pork belly fat iodine value (IV) and loin lean intramuscular fat (IMF) content in 158 pork belly primals and 419 loin chops. The calibration model revealed a 90.6% and 88.9% accuracy for the Tellspec NIR to predict saturated fatty acids (SFA) and IV, respectively, in the belly fat. The calibration model accuracy for the other belly fatty acids revealed an accuracy of 66.3–86.1%. Using the Tellspec NIR to predict loin lean IMF reported a lower accuracy for moisture (R^2^ = 60) and fat % (R^2^ = 40.4). This suggests that Tellspec NIR spectroscopy measures on the pork belly primal offers a cost-efficient, rapid, accurate, and non-invasive indicator of pork belly IV and could be used for the classification for specific markets.

## 1. Introduction

The demand, price, and market share of Canadian pork exports have evolved in recent years, resulting in increased export volumes and market diversity and variety. These changes have led to greater complexity in trade, mainly due to new and different market demands and access barriers [1]. The resulting increased competitive pressure in the international marketplace, where major pork exports compete over the same import markets, raises the importance of meeting importer quality specifications and product requirements to a new level. In addition, meeting consumer and market demands for consistent, predictable, and differentiated qualities offers an economic opportunity for the pork industry to optimize market returns across market segments [2]. Among other quality characteristics, fat composition, commonly assessed as iodine value (IV), as well as intramuscular fat (IMF) content in the loin muscle, are two of the main pork traits considered by both packers and buyers during a classification process. In the belly primal, currently the most expensive cut in North America, fat composition influences belly firmness and bacon yield [3]. On the other hand, IMF has a large impact on positive eating experience of cuts such as the loin [4], explaining the demand for highly marbled pork from both domestic and international markets.

Technology using near-infrared (NIR) spectroscopy has demonstrated its value as a non-invasive, accurate, and efficient tool in a variety of applications to assess meat composition and quality [5]. New low-cost NIR-based handheld devices are currently available, leading to opportunities to more easily assess important meat quality traits. Among these devices, the Tellspec spectrometer is advertised as an efficient alternative to evaluate food composition at a fraction of the price of traditional laboratory-based NIR systems. However, testing of novel devices on specific products is necessary in order to assess their performance under research and commercial conditions. The objective of this study was to assess the ability of the Tellspec handheld NIR device to predict fat IV and lean IMF content in pork primals (belly and loin, respectively).

## 2. Materials and Methods

### 2.1. Sample Collection and Preparation

A total of 158 pork belly primal cuts from a commercial abattoir were obtained for this study. The cranial, outer fat layer was scanned with the Tellspec NIR device and sampled for further fatty acid (FA) analysis.

A total of 419 loin chops from pigs commercially raised and slaughtered at the AAFC-Lacombe Research and Development Centre (Lacombe, AB, Canada) were used for the IMF analysis. Intact loin chops were scanned with the Tellspec NIR device and further analyzed as a comparison for moisture and fat content.

### 2.2. Tellspec Food Sensor Spectral Analysis

A clean-cut surface of the cranial outer fat layer of the pork belly primal and the loin chop was scanned to collect NIR spectra with a Tellspec Food Sensor device (Tellspec Inc., Toronto, ON, Canada) (device dimensions: 82.2 mm × 66 mm × 45 mm; weight: 136 g). The Tellspec Food Sensor device collected NIR spectra measures in reflectance mode (wavelength range of 900–1700 nm), with an evenly distributed spectral resolution (3 nm), and had a window size of 5 mm × 9 mm. The scans were taken by a single, uncooled InGaAs photodetector (1 mm), which is protected by a quarts lens. Samples were analyzed immediately after being removed from the refrigerator with a sample temperature of 2 °C. Using the “point mapping” measurement approach, spectral measurements were taken in four different locations per sample, then averaged.

### 2.3. Fatty Acid Analysis—Gas Chromatography and IV

After the collection of NIR spectra, a 5 g sample of the inner subcutaneous fat layer was sampled from each animal and stored at −80 °C until further FA analyses. Then, 50 mg sub-samples were obtained from the subcutaneous fat samples, which were then freeze-dried, direct methylated with 0.5 mol L-1 sodium methoxide, and then analyzed for FA methyl esters using gas chromatography. The gas chromatography methodology applied is described by Turner et al. [6], which used a Varian 3800 GC, equipped with an 8100 autosampler (Varian, Walnut Creek, CA, USA) with a 30 m Supelco (Bellefonte, PA, USA) SP-2340 capillary column (i.d. 25 μm). The conditions included a 20:1 split ratio, a temperature ramp staring at 50 °C for 30 s, up to 170 °C at 25 °C/min, held for 3 min, up to 180 °C at 2 °C/min, and, finally, up to 230 °C at 10 °C/min. The injector and flame ionization detector were set at 250 °C. The Nu-Check Prep Inc. GLC-463 and GLC-603 standards (Elysian, MN) were used to identify the individual FA based on retention time. Based on the analyzed FA composition, the IV of fat was calculated through the equation described by AOCS [7]: IV = [C16:1] × 0.95 + [C18:1] × 0.86 + [C18:2] × 1.732 + [C18:3] × 2.616 + [C20:1] × 0.785 + [C22:1] × 0.723 (brackets indicate the proportion of an individual FA (% of total FA)). 

### 2.4. Chemical Analysis

A mid-loin chop was cut from each loin and then finely comminuted using a Robot Coupe Blixir BX3 (Robot Coupe USA Inc., Ridgeland, MS, USA). A 50 g sub-sample of the resulting mid-loin chop grind was collected into a pre-labelled Whirl-Pak bag and frozen at −35 °C. The sub-sample was then thawed for 24 h prior to analyzing the moisture and fat content using the CEM rapid analyzer systems (Smart Turbo Moisture Analyzer Model 907,990, and Smart Trac Fat Analyzer Model 907,955 (CEM Corporation, Matthews, NC, USA)).

### 2.5. Statistical Analysis

Statistical analyses were conducted using SAS version 9.4 (SAS Institute Inc. Cary, NC, USA). One of every three samples were randomly selected for the validation set (n = 52 for belly fat; n = 139 for chop lean), with the remaining samples employed for the calibration set (n = 106 for belly fat, n = 279 for chop lean). Mathematical transformations were applied to the Tellspec NIR spectra prior to analysis to standardize and reduce error from light scattering effects, as well as to improve the accuracy of the prediction models. The Tellspec NIR spectra were subjected to multiple data transformation/pre-treatment methods; however, only transformations with the highest coefficient of determination (R^2^) have been reported (Table 1), including the Savitzky–Golay smoothing first (SG1st). A k-fold cross-validation (seven splits) was performed with partial least squares regression (PLSR) models within the calibration set. The R^2^ and root–mean–square error of cross-validation (RMSECV) from the resulting PLSR models were used to determine the calibration accuracy of each model. Factor selection was conducted by selecting the highest number of factors where the calculated T^2^ Hotelling statistic was significantly lower than with one fewer factors. The calibrated PLSR models were then applied to the validation set, and the model performances, R^2^, root–mean–square error cross-validation (RMSECV) and ratio of standard deviation to standard error of cross-validation (RDPCV) were reported. Additionally, the ratio of standard deviation (SD) was reported as a measure of prediction performance.

## 3. Results and Discussion

### 3.1. Descriptive Statistics

Table 2 summarizes the descriptive statistics for groups of analyzed FA and IV for all belly fat samples, as well as the compositional parameters for all loin chop lean samples. Variation was low for belly fat SFA, MUFA, IV, and chop lean moisture % (coefficient of variation (CV) < ~10%). Variation was intermediate for belly fat PUFA, n-3, n-6 (CV = 15–30%), and the greatest variation was observed in chop lean IMF (CV > 30%). The mean chemically extracted fat % of chop lean (IMF) observed in this study (IMF = 3.91%) was comparable to the mid-loin IMF percentage calculated by chemical extraction (IMF = 2.70% IMF) in a prior study by Uttaro et al. [8] Additionally, a study assessing IMF in various locations in pork loins of different sex and sire lines reported a range of 73.40–74.33% moisture in loin chop samples across sex and breed lines and a range of 2.77–4.02% extractable lipid of loin chops in the same animals, which are also comparable to the results reported in the current study [9]. The belly fat calculated IV in this study (mean IV = 68.0) was comparable to the belly fat IV calculated in a prior study (n = 52.5) by Lam et al. [10] using fat from the outer layer of the posterior belly subcutaneous fat, as well as a study assessing pork subcutaneous fat of belly, ham, loin, and shoulder cuts (n = 162) was similar (mean IV = 68.54) [11]. The belly fat FA values measured in this study were similar to FA values in the belly primal, reported in a prior study by Harris et al. [12] that assessed dietary treatments across breeds (n = 192).

### 3.2. Calibration and Validation Analysis

The rising demand in achieving high quality pork primals, which is driven by the increase in economic value and consumer preferences, has led to the need for portable, direct, non-invasive, and high-throughput methods for pork quality assessment. Substantial research exists studying NIR spectroscopy methods to measure meat or carcass quality attributes. More specifically, NIR methods have demonstrated effectiveness in measuring chemical composition attributes to predict quality for classification [5].

A prior study tested the Tellspec NIR scanner as a tool to differentiate between different halal–meat species, showing high accuracy of prediction for intact meat of lamb (R^2^ = 88–100), beef (R^2^ = 93.6–100), chicken (R^2^ = 97.3–100), and pork (R^2^ = 100) [13]. Other studies have evaluated the performance of the Tellspec NIR device on processed pork samples to predict IMF [14] and on beef tenderloin and sirloin samples to predict aging [15]; however, the Tellspec NIR device performance has yet to be tested for predicting intact pork belly fat IV and loin IMF content, which is reported in the current study.

The Tellspec NIR belly prediction accuracies (R^2^) within the calibration set ranged from 66.3 to 90.6% for the fat parameters and 88.9% for IV (Table 1). Additionally, using the R^2^ validation model to predict SFA and IV in belly had the highest validation accuracy (R^2^ = 77.2; R^2^ = 87.1, respectively) compared to all other FA parameter predictions. A former study by Prieto et al. [16] assessed the potential of NIR spectroscopy to predict FA composition and IV in subcutaneous fat in the pork shoulder, which observed a high IV prediction accuracy of the calibration model of R^2^ = 0.95 [16]. The higher IV prediction accuracy reported in the study by Prieto et al. [16] may be due to their use of a hand-held ASD fiber optics reflectance probe inserted into the shoulder, in which the probe method allows for 10 s of equipment scanning, while the operator moved the probe across the fat surface, resulting in NIR spectra data for a larger subcutaneous fat area (approximately 10 cm^2^).

When observing the expected variation between the IV and Tellspec NIR measure of belly samples (RMSECV), the RMSECV ranged from 0.08 to 1.86 (Table 1). This demonstrates that the Tellspec-predicted IV of belly samples would be correct to within 0.16 to 3.72 IV (2 × RMSECV) for 95% of the samples when using the calibration model. This is sufficient accuracy for online classification at a pork processing plant.

Figure 1 illustrates the raw near-infrared spectra for all pork belly fat samples (Figure 1a) and all loin chop lean samples (Figure 1b). Regarding the loin chop lean spectra in Figure 1b, we observe two absorbance bands, which occurred at similar wavelength ranges in a study by Prieto et al. [17] that reported example raw visible NIR spectra of intact meat samples of several species, including pork, using a portable NIR instrument. We see a large absorbance band around 1400 nm in Figure 1a,b, which may be due to the absorbance of water, which is reasonable due to the high water content of fat and lean tissue (65% moisture in chop lean samples; Table 1). In addition, the absorbance band observed around 1700 nm in Figure 1a,b corresponds to the NIR absorption bands around 1600–1800 nm that are due to straight carbon chain fatty acids [18]. NIR absorbance bands of fatty acids beyond 1692 nm may exist, but they are not visible in the plotted range seen in Figure 1a.

Currently, rapid IV prediction using commercially available devices, such as the NitFom^TM^ probe, is used to classify carcasses, with a low IV indicating firmer fat (higher quality) and a high IV indicating softer fat (lower quality). These penetrative systems have been developed to work on whole carcasses, averaging layers of fat, not as non-invasive systems on individual primals and specific layers of fat, and their cost and installation requirements have limited their adoption by the pork sector. As the prediction accuracy of the Tellspec NIR to predict IV was above 77.2% (Table 1), this may suggest that the Tellspec NIR device could serve as an alternative to assess fat quality in a cost-effective, efficient, and non-invasive way. With firmness and FA composition being the most important quality attributes of the belly cut primal, this may suggest an opportunity for using the Tellspec NIR device to individually classify the belly cut primal for quality.

High IMF is a desirable attribute associated with high quality assessment of loin chops, which is commonly subjectively classified by packers using marbling standards. Therefore, loin chop IMF is a major factor in pork production profitability. Currently, chop IMF is assessed subjectively by packers using score cards on the processing line. As previously reviewed by Bohrer and Boler [19], subjective pork quality standard scoring, including marbling evaluated using the National Pork Producers Council (NPPC) [20] marbling standards, revealed a moderate correlation (r = 0.48) with IMF %. Despite the relatively weak relationship, marbling assessed subjectively is one of the main attributes considered by industry to classify loin, emphasizing the need for alternative technologies to accurately predict whole cut IMF %.

A prior study assessed loin IMF content using NIR hyperspectral imaging of the rib end, revealing a model prediction accuracy of R^2^ ≥ 0.89 [21]. Higher prediction accuracies observed in the study by Huang et al. [21] may be due to the advantage of hyperspectral imaging over traditional NIR spectroscopy, with its ability to provide NIR spectral data across a set of images, resulting in multiple benefits such as chemical composition identification in non-homogeneous samples [22].

The RMSECV was 0.82 and 0.72 for moisture and fat %, respectively, suggesting that the Tellspec-predicted moisture and fat % of loin samples would be correct within 1.64 and 1.44 fat and moisture %, respectively, for 95% of the samples (Table 1). Despite the observed reasonable prediction variability (RMSECV) using the calibration data, a low accuracy in prediction was observed for loin chop moisture (R^2^ = 60) and fat % (R^2^ = 40.4) for the validation set. This may suggest other factors must be considered when assessing loin IMF using NIR spectroscopy methods. An element of portable NIR spectroscopy devices that should be considered is the optic window size, which affects the surface area, in which the infrared light beam can pass through to penetrate the sample [23]. While devices with small windows may be suitable for homogenized samples, such as ground meat, the same device may not collect representative data for intact muscle IMF content.

The “point mapping” measurement approach used in this study should also be considered. Point mapping measures multiple different spatially unrelated areas of the cut surface and are analyzed consecutively [24]. An alternative method may involve area mapping, which requires the sampling of larger areas to define a series of spectra and provide a two-dimensional representation, which may be more suitable to accurately measure IMF, but would require more labor and time.

Overall, this study demonstrates that the Tellspec NIR device is an effective and economical option to be used as a predictor for belly fat IV. In contrast, the Tellspec NIR device is not as effective in predicting loin IMF. As NIR spectroscopy technologies continue to advance, opportunities to integrate these tools into existing fast-paced decision-based classification frameworks in processing plants will become more possible, allowing for optimum downstream processing decisions for higher profit margins.

## Figures and Tables

**Figure 1 foods-12-01629-f001:**
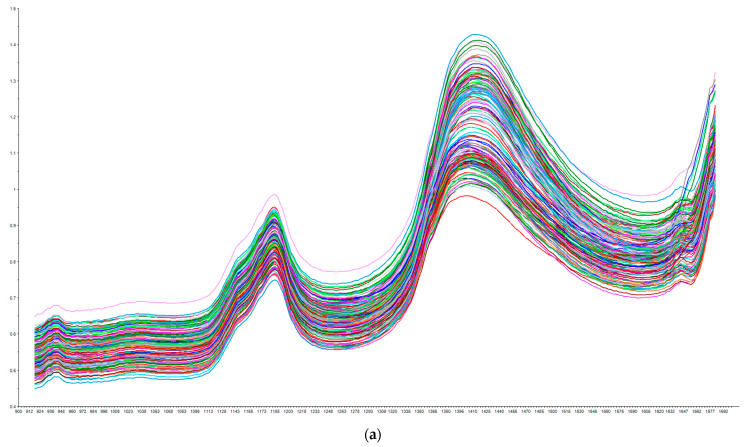
(**a**) Raw near-infrared spectra collected on all pork belly fat samples (N = 158) using the Tellspec NIR device. X axis = Absorbance; Y axis = Wavelength (nm). (**b**) Raw near-infrared spectra collected on all loin chop lean samples (N = 419) using the Tellspec NIR device. X axis = Absorbance; Y axis = Wavelength (nm).

**Table 1 foods-12-01629-t001:** Prediction parameters (Tellspec NIR on the pork belly primal fat samples and the loin chop lean samples) for fat compositional measures in belly fat and fat and moisture content in chop.

	Belly Fat						Chop Lean	
	SFA	MUFA	PUFA	n-3	n-6	IV	Moisture	Fat
Derivative	SG1^st^	SG1^st^	SG1^st^	SG1^st^	SG1^st^	SG1^st^	Original	Original
Minfac	7	8	7	6	7	7	8	7
PRESS	0.36	0.49	0.60	0.69	0.59	0.39	0.65	0.80
RMSECV	1.38	1.86	1.47	0.08	1.39	1.68	0.82	0.72
RPDCV	3.01	2.44	1.95	1.61	1.96	2.76	1.53	1.26
SD	4.17	4.54	2.86	0.13	2.73	4.66	1.26	0.91
R^2^ calibration	90.6	86.1	77.7	66.3	78.0	88.9	60.0	40.4
R^2^ validation	77.2	53.7	58.7	51.6	58.8	87.1	58.6	34.3

Note: SFA, saturated fatty acids, MUFA, monounsaturated fatty acids; PUFA, polyunsaturated fatty acids, n-3, omega-3 fatty acids; n-6, omega-6 fatty acids; IV, Iodine Value; Moisture and Fat are percent totals; SG1^st^: Savitzky–Golay smoothing first; Minfac, minimum factor; PRESS, predicted residual error sum of squares, RMSECV, root-mean square error of cross-validation; RDPCV, ratio of standard deviation to standard error of cross-validation; SD, standard deviation.

**Table 2 foods-12-01629-t002:** Descriptive statistics for fatty acid parameters, iodine value of belly fat samples (n = 158), and moisture and fat composition of chop lean samples (n = 418) for calibration and validation sets.

	Belly Fat						Chop Lean	
	SFA	MUFA	PUFA	n-3	n-6	IV	Moisture %	Fat %
**Calibration Set**								
n	105	105	105	105	105	105	105	279
Mean	37.6	43.8	18.6	0.92	17.7	19.3	68.0	73.4
SD	30.3	34.0	10.4	0.50	9.91	17.1	57.6	70.1
Min	4.17	4.54	2.86	0.13	2.73	0.93	4.66	0.91
Max	45.9	50.3	25.1	1.23	23.9	22.3	79.4	75.4
CV	11.1	10.4	15.3	14.5	15.4	4.82	6.85	1.24
**Validation Set**								
n	52	52	52	52	52	52	52	139
Mean	37.6	44.2	18.2	0.89	17.3	19.4	67.6	73.4
SD	31.4	35.8	11.9	0.58	11.3	17.7	61.3	70.7
Min	3.97	4.37	3.01	0.15	2.87	0.91	4.67	0.88
Max	44.6	51.5	25.0	1.25	23.7	22.4	78.4	75.2
CV	10.5	9.89	16.6	16.4	16.6	4.72	6.91	1.20

Note: SFA, saturated fatty acids, MUFA, monounsaturated fatty acids; PUFA, polyunsaturated fatty acids, n-3, omega-3 fatty acids; n-6, omega-6 fatty acids; IV, Iodine Value; SD, standard deviation; CV: Coefficient of variation.

## Data Availability

Not applicable.

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
