# Peer review of "Performance of a Handheld Near-Infrared Spectroscopy Device to Predict Pork Primal Belly Fat Iodine Value and Loin Lean Intramuscular Fat Content"

_foods, 2023, doi:10.3390/foods12081629_

Round 1

Reviewer 1 Report

Fast, accurate, and nondestructive methods for the estimation of meat quality are important for producers and consumers. The manuscript presents an interesting study related to the use of a portable low-cost spectral instrument to predict pork quality. The manuscript needs to be improved, in particular with regard to a better description of the results.

Note to the manuscript:

Title of Fig. 1 and 2 – “Raw visible and near-infrared spectra ……”. The used spectral instrument work in the near-infrared region from 900 to 1700 nm. Why the visible range is mentioned? The same is in the text.

Fig. 1 and 2 - Raw spectra presented in Figs 1 and 2 are not very informative. It would be good to present the spectra transformed as the second derivative. Then the maximums and their position would be clearly visible.

Why table 2 in the text is before table 1?

The investigated samples are divided into a calibration set and an independent validation set. No descriptive statistics are presented for the calibration and the validation set. The values of R2Validation are presented in Table 2, but the values of errors of determination of examined parameters for the verification group are not presented. 

The great differences in the accuracy of the determination of some of the parameters for the calibration and validation set have not been explained.

A good idea of the accuracy of the determination would be obtained by figures, presenting the relationship between actual and NIRS predicted values. Such graphics are not presented in the manuscript.

Author Response

Please see attachement.

Reviewer 2 Report

The manuscript contains interesting new information. I suggest a bit more review and rewording to make the measurement results more interpretable.

line 13 Explanation of SFA abbreviation missing

Chapter 2.1. The fat content of the samples, its distribution, is a problem in the case of meat or loin slices. I don't feel it is sufficient that the samples were scanned at four points. How was it verified that this allowed the fat distribution for the whole sample to be examined?

Chapter 2.3. The gas chromatography method used to determine the fatty acid profile is incomplete. I am missing the type of apparatus and the measurement parameters

lines 92-93: root-mean-square error (RMSE) of cross-validation - the valid abbreviation is RMSECV

Table 2 it is useful to include the ranges of measurement, because only then can the value of RMSECV be interpreted

The numbering of Tables 1 and 2 is incorrect.

In Tables 1 and 2 the terms moisture and fat content are not always % (e.g. sample number), this should be corrected.

The two tables should be edited into one table to make the data easier to understand

Lines 154-158 That's exactly the problem I saw with the spectral recording. With four correct scans of such an inhomogeneous sample, it is not possible to get a realistic spectrum

Fig.1 - I suggest renaming the x and y axes in the figure and rescaling. If fewer numbers are included, it is easier to see the values. I miss the comparison of the two spectra. Describe in a few sentences the reason for the difference

line 201 Explanation of NPPC abbreviation missing

lines 221-223 Another element is the spectroscopy model, in which the Tellspec NIR technology uses reflectance (IR beam reflects from the sample surface), rather than transmittance (IR beam passes through the sample) I disagree with the statement.

Transmittance spectra can be recorded over a narrower wavelength range than reflectance spectra. It follows that their evaluation is also more problematic.

Author Response

Please see attachement.

Reviewer 3 Report

Introduction part is very short, author can add the reason and importance of IV in this part. Other non destructive methods may also be discussed. Chemometric analysis may be elaborated in introduction portion. 

I table 2 is placed before table 1 in the paper. 

Results and discussions: Only PLSR has been applied on data, PCA plot might be used to see the distribution of data and to explore similarities and differences.

Add a figure with average of both kind of spectra in order to see differences in peaks

Round 2

Reviewer 1 Report

The manuscript is improved but still needed correction.

Title of Fig. 1 a and b – “Raw visible and near-infrared spectra ……”. The visible range is from 380 to 760 nm. The used spectral instrument work in the near-infrared region from 900 to 1700 nm, not in the visible range. Remove “visible” in the title of Fig. 1. The same is in the text – line 173.

The place of table 1 is in the “Results and discussion” section, after the table with the descriptive statistics of the calibration and validation data set.

Table 2 – Line Max – the max values are much smaller than the min values. There is probably a line mix-up. Check values in Table 2.

Author Response

Thank you for your feedback for Round 2.

Reviewer 2 Report

The authors have corrected the errors.

Author Response

Thank you for your feedback.

Reviewer 3 Report

Corrections have been made in the revised version.

Author Response

Thank you for your feedback.